# A new CT scan methodology to characterize small aggregation gravel clast contained in soft sediment matrix

Laurent Fouinat[1], Pierre Sabatier[1], Jérôme Poulenard[1], Jean-Louis Reyss[2], Xavier Montet[3], Fabien Arnaud[1].

[1]EDYTEM, Université Savoie Mont Blanc, CNRS 73376 Le Bourget du Lac Cedex, France
[2]LSCE, Université de Versailles Saint-Quentin CEA-CNRS, avenue de la Terrasse, 91198 Gif-sur-Yvette cedex, France
[3]University of Geneva Department of Radiology and Medical Informatics Genève, Rue Gabrielle-Perret-Gentil 4, CH-1211, Switzerland

*Correspondence to*: Laurent Fouinat (laurent.fouinat@univ-smb.fr)

**Abstract.** In recent years, developments of X-ray techniques such as the CT scan analysis have been used in the geosciences community. CT scanning is a fast, non-destructive method allowing the assessment of relative density of clasts in natural archives samples. This study focuses on the use of this method to explore instantaneous deposits as major contributors to sedimentation of high elevation lakes in the Alps such as Lake Lauvitel system (western French Alps) featuring steep slopes, strong torrential activity and gullies directed towards the lake. Instantaneous deposits identified in the sediment core were eighteen turbidites and fifteen layers of poorly sorted fine sediment associated to presence of gravels revealed with our methodology since 1880 AD. These deposits are respectively interpreted as flood and wet avalanches induced, based on underwater mechanisms found in the literature and comparison with historical as well as natural archives records. This CT scan approach is suitable for instantaneous deposit identification to reconstruct past evolution and may be applicable a wider variety of sedimentary archives alongside existing approaches.

## 1 Introduction

Over the last 50 years, X-ray radiographs were initially used to explore the internal structure of sediment cores (Bouma, 1964; Baker and Friedman, 1969) in order to optimize the opening process or even explore bioturbation structures in the sediment (Howard, 1968). One of the technical problems was the loss of information with respect to depth, as the radiographs are a plane representation of a 3D structure. A recent review of CT scans in the geosciences (Cnudde and Boone, 2013) demonstrates the growing application possibilities of X-ray technology as well as the limits of the technique. Improvements in CT scanning allowed exploration of complex sedimentary structures through 3D reconstructions, leading to improvement compared to classic 2D imaging (Pirlet et al., 2010; Bendle et al., 2015). The method is based on the relative density of each voxel constituting the chosen sample. The position of each voxel is set on a x, y, z frame allowing association of adjacent identical density voxels to identify sediment constituents. Image analysis of the 3D numerical model can then be used to obtain a quantitative information about selected constituents as well as a volumetric information (Bolte and Cordelieres, 2006). This type of methodology was recently used to identify and quantify gypsum formation in marine sediments (Pirlet et al., 2010) as well as different sediment clast deposition in a glacio lacustrine varved context (Bendle et al., 2015).

High elevation lake situated in mountain areas are often characterized by elevated and highly variable sedimentation rates (Arnaud et al., 2016). The variety of erosion processes caused by chemical and mechanical weathering as well as rock breaking by frost action creates heterogeneous grain size elements. Extreme climatological events can trigger several high energy transport mechanisms which could induce deposition of these elements in lake sediments. Depending on the processes, extreme events may induce different sedimentary structures containing coarse grains (Arnaud et al., 2002; Sletten et al., 2003; Nielsen et al., 2016). Fluvial events such as floods are known to be able to transport very large quantities of sediment in a short period of time (Sturm and Matter, 1978; Jenny et al., 2014). In recent years, they were also largely identified as major sedimentary income in high elevation lakes (Giguet-Covex et al., 2012; Glur et al., 2013; Wilhelm et al., 2013a; Wirth et al., 2013; Wilhelm et al., 2015). As floods are formed by heavy precipitations, the torrential stream transporting sediment in suspension will enter into the lake and create a

density current resulting in a characteristic deposit called turbidite (Gilli et al., 2013). The density difference between subaerial flow and lake water can create different underwater flow, but each type will result in a coarse grains base with a fining upward trend (Sturm and Matter, 1978). In certain cases, lake surroundings may include gullies orienting subaerial flow into the water. Two mass-wasting types of transport related to these gullies were identified in high elevation lakes sediments. First ones are debris flow triggered by water transport but with a lower water content compared to floods (Postma, 1986; Dasgupta, 2003). Their transport capacity is thus increased and they will form specific deposits in underwater environments due to their higher density and sediment cohesion (Mulder and Alexander, 2001). Typical deposits are composed of a load cast layer most of the time with basal erosive surface, superimposed by a fining upward trend layer constituted of the finer sediment fractions deposition (Sletten et al., 2003; Irmler et al., 2006). The second type of transport can be attributed to wet avalanches occurring in spring time requiring sufficiently steep slopes to occur, typically greater than 28° but have been observed on slopes as low as 15° (Jomelli and Bertran, 2001; Ancey and Bain, 2015). Wet avalanches have carrying capacities going from fine eolian sediment deposited on snow to cobbles or boulders (van Steijn et al., 1995; Blikra and Nemec, 1998; Jomelli et al., 2007; Sæmundsson et al., 2008; Van Steijn, 2011). The main transport mechanism is the flow of water saturated snow carrying sediment, which can be deposited directly into lake water or on frozen lake surface (Luckman, 1975, 1977). Wet avalanches in lacustrine deposits have been identified by Vasskog et al., (2011) using grain size analysis to identify layers of poorly sorted grains accumulation with the addition of gravel presence resulting in a multi modal grain size distribution. Wet avalanche deposits are characterized by low underwater flows; the lake water snow being denser as compared to snow, the resulting deposits would be formed as poorly sorted grain accumulation. In the case of snow falling on the lake ice, the sediment accumulated would then be randomly distributed at thaw season as drop stones in lake sediment (Luckman, 1975). Moreover, terrestrial organic matter (OM) is sometimes associated with these deposits (Irmler et al., 2006; Wilhelm et al., 2013b; Korup and Rixen, 2014).

All of these high energy processes induced the presence of coarse grained deposits, and methods used to identify and count the coarser elements have been based on wet sieving successive layers of sedimentary cores which is both time consuming and destructive method (Seierstad et al., 2002; Sletten et al., 2003; Nesje et al., 2007; Vasskog et al., 2011). In this study, we propose a complementary method to grain size analysis to better characterize these coarse grains in a simpler, faster and non-destructive way based on the use of CT scanning. Moreover, this technique is suitable for identifying sedimentary structures in a 3D numerical model, allowing other perspectives in sedimentary studies. We applied this method in the on sediment cores from Lake Lauvitel located in the Oisans valley (western French Alps). This site has the advantage of presenting multiple high energy events with the presence of steep slopes and three avalanche corridors directed towards the lake.

## 2 Materials and methods

### 1.1 Study site

Lake Lauvitel (44° 58' 11.4"N, 6° 03' 50.5" E) is located 1500 m above sea level (a.s.l.) in the Oisans valley of the western French Alps, 35 km southeast of Grenoble. The lake covers an area of 0.35 km² and is 61 m deep, and the total drainage area is approximately 15.1 km². The lake was created after a large rockslide dated to 4.7±0.4 kyr [10]Be exposure age (Delunel et al., 2010). The natural permeable dam created after this event caused a change in lake level of approximately 20 m. Due to geomorphological settings, slopes around the lake are very steep and three avalanche corridors (C1, C2, and C3) are present on the western side of the lake (Fig. 1b). They are accompanied by the presence of snow accumulation at their bottom in spring (National Park ranger, Jérôme Forêt, pers. comm.), and avalanches have been observed in C1 (Fig. 1e). The watershed bedrock consists mainly of granite and gneiss, with minor outcrops of sedimentary rocks (Triassic limestone). The C1 track ends in an upper basin in the northern part of the lake, likely with no connection to the deeper part of the lake. C2 and C3 are located just above the coring location; there is no clear evidence of an obstacle preventing the sediment input from reaching the coring location. From the end of December to the beginning of May, the lake surface is frozen, and snow covers most of the watershed. The lake and its surroundings are situated in the Ecrins National Park restricted area.

Figure 1

### 1.2 Core description and methods

The core LAU11P2 (76 cm) was retrieved using a short UWITEC gravity corer to obtain a well-preserved interface, and LAU1104A (104.5 cm) was retrieved using a piston corer with a 90-mm sampling tube at the same location. The cores were split lengthwise and photographed at high resolution (20 pixels mm$^{-1}$). We examined in detail the visual macroscopic features of each core to define the different sedimentary facies to determine the stratigraphic correlation between the two cores.

CT scanning was performed at Hopitaux Universitaires de Genève (HUG) using a multidetector CT scanner (Discovery 750 HD, GE Healthcare, Milwaukee, Wis). The acquisition parameters were set as follows: 0.6-s gantry rotation time, 100 kVp, 0.984:1 beam pitch, 40-mm table feed per gantry rotation, and a z-axis tube current modulation with a noise index (NI) of 28 (min/max mA, 100/500) and a 64×0.625-mm detector configuration. All CT acquisitions were reconstructed with the soft tissue and bone kernel in order to enhance the density contrast (Tins, 2010). The images reconstructed with the bone kernel were used for subsequent analysis. The raw DICOMM images were converted to an 8-bit .TIFF format using Weasis (v2.0.3) viewer. The radiograph resolution is 512x512 pixels, with up to 256 grey scale values. In this study, the sediment core was divided into 1,045 1-mm-thick frames, each pixel corresponding to a resolution of up to 500x500 μm and thus a voxel of 0.25 mm$^3$. The images were then stacked using the Image J FIJI application, and image treatments were performed using the 3D Object Counter plugin (Bolte and Cordelieres, 2006). First, we set a threshold to isolate the selected grey values, and we then applied a despeckle filter to remove the noise due to measurement. Finally 3D Object counter was used to reconstruct the particles and characterize them in a 3D coordinate system.

Grain size measurements were carried out on the core using a Malvern Mastersizer 800 particle-sizer at a lithology dependent sampling interval. Utrasonics were used to dissociate particles and to avoid flocculation. Several layers of gravel-sized mineralogic particles were identified (Fig. 2a) in the LAU1104 sediment core. To obtain a quantitative estimate of these particles, we passed samples through a 1-mm mesh and wet-sieved the sediment at variable intervals from 1 to 3 cm depending on the gravel concentration. The number of particles >2 mm and macro-remains present in the sieve were counted for each interval in the core LAU1104A.

The chronology of the Lake Lauvitel sediment sequence is based on short-lived radionuclide measurements. The short-lived radionuclides in the upper 75 cm of core LAU11P2 were measured using high-efficiency, very low-background, well-type Ge detectors at the Modane Underground Laboratory (LSM) (Reyss et al., 1995). The sampling intervals followed facies boundaries, resulting in a non-regular sampling of approximately 1 cm. Twelve thick beds (at depths of 10.4-12.7, 17.3-19, 22.9-24.8, 29.7-30.9, 38-39, 40.6-42.4, 43.1-44.2, 45.7-50, 54.5-56.9, 60.4-62.5, 64.1-66 and 67.2-68.3 cm) were not analyzed because they were considered to be instantaneous deposits or part of an instantaneous deposit (see Results). $^{210}$Pb excess was calculated as the difference between total $^{210}$Pb and $^{226}$Ra activities.

## 3 Results

### 3.1 Lithostratigraphy

The core lithology is composed of three facies (Fig. 2a). Facies 1 (F1) is silty-clay, dark-brown, finely laminated layer. It is interbedded by two other facies that are almost always associated with each other: Facies 2 (F2) is a normally graded bed from coarse sand to silt, sometimes with an erosive base; this facies is always associated with a thin white clay-rich layer Facies 3 (F3) on the top. Fig. 2b presents typical normally graded beds with grain size distribution (in red) characterized by a median grain size (Q50) of 44.1 µm and a mode of 81 µm. F1 (in green) exhibits a median grain size of 13.5 µm and a mode of 11.9 µm. Sometimes, F1 presence coincides with coarse gravel in the sediment, then the median grain size is similar 9.7 µm, but two modes are discernible at 7.2 and 258 µm. Sorting parameter reveals different values depending on the deposit type; 2.50 average in the normally graded beds, 2.65 for the annual sedimentation and 3.05 for annual sedimentation with gravel presence. The small Q50 difference between annual sedimentation with and without gravel supposes limited addition in the fine grains fractions. Meanwhile, fraction over 100 µm and bad sorting and Q90 reveal a significant addition of sand size grains in the gravel layers. The presence of terrestrial macro-remains is sometimes identifiable in F2. A total of 18 normally graded beds are present in the core LAU1104A, with thicknesses ranging from 0.7 to 13 cm. We also identified 15 layers with poorly sorted fine sediment associated to gravel presence, with thicknesses ranging from 0.3 to 5.9 cm.

The CT scan analysis is based on relative density expressed on the histogram (Fig. 3a) representing the frequency of each of 1-255 levels of grey (0 is not shown on the graph due to overrepresentation corresponding to the background signal). Three modes representing the most frequent values are apparent in the histogram which would be associated with certain types of sediment. The first mode is centered on the 106 value. After selecting this mode, we isolated the numerical values in order to map them by using the plugin. The corresponding elements in the sediment core were

small OM macroremains such as a pinus twig found at 58 cm of depth (Fig. 3-e1). We thus selected the 95-125 range to identify OM. The second mode, centered on the 174 value, is relatively denser than OM. Its larger spectrum and high count values correspond to the most common element in the sediment core, which would be the silty clay sedimentation matrix (Fig. 3b). The last mode is essentially the 255 level of grey. It is the densest value possible, thus corresponding to denser elements present in the silty-clay matrix. We selected the 250-255 value range and isolated them, and searched for corresponding particles in the sediment core. Wet sieving allowed identification of gravel-sized granite elements in the sediment core (Fig. 3-e3-e4).

To compare objects counted numerically and objects counted manually, we need to know the size limit in units of volume (voxels), which is equivalent to 2-mm-diameter holes in a sieve. In 2D, a particle is retained in the sieve only if at least two sides are 2 mm in length, meaning at least two sides are 4 pixels long. Therefore, a particle of 16 (4x4) pixels with four sides that are 2 mm long will be retained in the sieve. However, if the same particle is missing 1 corner (minus three pixels, corresponding to a particle of 13 pixels), the particle would still be large enough to be retained in the sieve. This angular shape is more likely to be encountered in avalanche deposits. Consequently, we set the size limit of the 3D Object Counter plugin to 13 pixels, which corresponds to 13 voxels. The organic macroremains are composed of herbs, twigs or even roots, and their shapes were very complicated. Therefore, we did not choose any volume limit in their identification process.

In the LAU1104A sediment core, a total of 456 gravel clasts equal to or larger than 13 voxels were identified for a total of 112 683 mm$^3$. The largest high-density object recovered from the core LAU1104A was an angular piece of granite of over six centimeters on its longest side and weighing 206.03 g. Considering a volumetric mass density of 2.7 t/m$^3$, its volume can be estimated to be 76 307 mm$^3$ ($\rho$=m/V). In comparison, the numerical volume is estimated to be 376,187 voxels, corresponding to 89,690 mm$^3$. A difference of +15% in the volume for the CT counting is observed probably due to pixel resolution. The volume is slightly overestimated, but still close to the actual rock volume.

We then compared the 3D Object Counter results and the coarse grains recovered from the sediment cores in slices of variable thickness ranging from 1 to 3 cm. The depth 97-98 cm had no gravel > 2 mm in either the manual or numerical counting (Fig. 3b, d). When considering a large amount of gravel, the manual and numerical counting methods showed differences. For depths 15-18, 42-44, 44-46, 51-52, and 72-73 cm, the number of gravel clasts was always underestimated by the numerical counting. As the 3D Object Counter plugin is identifying objects from one pixel and its 8 neighbours in 2D and its 26 neighbours in 3D (Bolte and Cordelieres, 2006), the identification of objects could vary especially because of the noise treatment and when the object size is close to the image resolution. The numerical counting result is slightly underestimated compared to the manual counting result (30% on average). On the contrary, depths 5-7 and 46-48 cm showed an overestimation by the numerical counting (77% on average). Considering the resolution, it is possible that a certain number of aggregated sand grains could have been considered gravel by the numerical counting method, leading to an overestimation. This could be explained by the presence of flood deposits in these two depths (Fig. 3b). Aggregated sand-sized elements would be considered by numerical counting as larger elements. In addition, the sand-sized elements are rounder and would go through the sieve, as opposed to an angular particle of similar volume which would be retained in the sieve. Overall, from this comparison

between the numerical and the manual counting and accounting for the previously mentioned CT scan bias, we obtained a relatively well-constrained positive correlation (r=0.79, n=8; p-value=0.0077) (Fig. 3d).

The OM counting identified 7,413 objects, spread throughout almost every part of the sediment core. The largest OM element found in the core was 6,949 voxels in size, corresponding to 1,732 mm$^3$. This OM element was situated at a depth of 58 cm in the middle of a flood deposit (Fig. 3b) and was identified as a pinus tree twig (Fig. 3e-1). In total, 89.2% of the numerically counted OM elements are under 3.25 mm$^3$ (13 voxels), and almost every element recovered in the sieve corresponded to small leafs, roots, twigs or herb macroremains (Fig. 3e-2).

Figure 2

### 3.3 Chronology

The $^{210}$Pb excess profile (Fig. 4) showed a regular decrease punctuated by drops in $^{210}$Pb$_{ex}$ activities. Following
(Arnaud et al., 2002), these low values of $^{210}$Pb$_{ex}$ were excluded to construct a synthetic sedimentary record, because these values are related to F2/F3 facies association, which is considered to be instantaneous turbidite deposits. Plotting on a logarithmic scale, the $^{210}$Pb$_{ex}$ activities revealed a linear trend (Wilhelm et al., 2012b). Applying the CFCS model (Goldberg, 1963), we obtain a mean accumulation rate of 3.7 +/- 0.3 mm yr$^{-1}$. The uncertainty in the sedimentation rate was derived from the standard error of the CFCS model linear regression. Ages were then
calculated using the CFCS model applied to the original sediment sequence to provide a continuous age-depth relationship. In addition, $^{137}$Cs and $^{241}$Am activity profiles present two peaks and one peak, respectively. The older peak in $^{137}$Cs activity at 28.1 cm is contemporary with the peak in $^{241}$Am activity, allowing us to associate it to the peak of nuclear weapons testing in the northern hemisphere in 1963 AD. The younger peak in $^{137}$Cs activity at 17.3 cm can be attributed to fallout from the Chernobyl accident in 1986 AD (Appleby et al., 1991). These two artificial
peaks are in good agreement with the CFCS model (Fig. 4). In addition, we compared the historical flood calendar from the Vénéon river valley from the RTM-ONF data base (http://rtm-onf.ifn.fr/) to the instantaneous deposits recovered from the lake sediment for the last 100 years. In local archives, eight major flood events occurred in 2008, 2003, 1987, 1962, 1955, 1938, 1922 and 1914 AD, could be correlated to the most important and recent graded deposits at depths of 0.4-2.9, 9.9-11.4, 18.7-20.1, 28.5-32.9, 38.2-39.6, 46-61, 64.9-66.7, and 67.7-69.1 cm,
respectively. The good agreement between these independent chronological markers and the $^{210}$Pb$_{ex}$ ages strongly supports our age-depth model for the last century and validates our interpretation that the F2/F3 facies correspond to instantaneous flood deposits.

Figure 3

**4 Discussion**

The Lake Lauvitel sedimentary record allowed identification of event related layers such as the normally graded beds. Median grain size (Q50) and the coarser 10$^{th}$ percentile (Q90) parameters within graded beds allows us to consider them to be turbidites caused by heavy rainfall in the watershed (Støren et al., 2010; Giguet-Covex et al., 2012; Wilhelm et al., 2012b, 2012bb, 2013a; Gilli et al., 2013; Wilhelm et al., 2015). In addition, gravels were evidenced in the upper part of the flood deposit. This location within the turbidite corresponds to receding torrential activity (Gilli et al., 2013). They are unlikely to have originated from the torrential activity due to the distance from the delta. The presence of gravel in the turbidites could also be attributed to debris flow activity resulting in an dense cohesive underflow transforming in a tubidite layer (Weirich, 1988). However, we do not observe a load clast at the base of the deposit as a typical debris flow would exhibit, but instead our results show sparse gravel presence in the upper part of the deposits (Fig. 2). Gravels within flood deposits could be linked to temporary tributaries only active during heavy precipitations, such as the avalanches corridors during summer season. We observed, in the homogeneous fine annual sedimentation, a similar scarce presence of gravel elements (Fig. 2). In those layers, the sorting does not seem to be different than annual sedimentation without gravel. This could be explained by the gravel elements falling directly in the lake water or on the frozen lake surface then producing drop stones.

Our results allowed identification of fifteen layers characterized by presence of numerous gravel elements accompanied by poorly sorted fine grains exhibiting multi modal grain size distribution (Fig. 2). This feature was observed essentially in several lakes in Norway and attributed to avalanche induced layers (Blikra and Nemec, 1998; Seierstad et al., 2002; Nesje et al., 2007; Vasskog et al., 2011). Wet avalanches occur at spring season when the snow pack is becoming unstable due to loss of cohesion in the structure caused by warmer temperatures. Spring is also thaw season for lake ice, hence, the avalanches could either be deposited on ice or enter directly in the water, as observed during the May 1$^{st}$ 2015 avalanche (Fig. 1). At that time, the snow flow originated from the C1 corridor in the northern part of the lake where an upper basin is present. It is likely to have no sedimentary connection with the deeper basin where the coring point is located. Snow avalanche detrital material can be integrated into lacustrine sediments in two ways. In the case of a frozen lake, surface avalanche deposits are spread across the ice and subsequently drop to the lake sediment from drifting ice. When an avalanche occurs while the lake is ice-free, the avalanche deposits directly enter into the water, and particles are concentrated in a more restricted area closer to the avalanche corridor. Fine sediment in between gravels could thus be originating from the lake bottom or/and from the avalanche and are consequently considered as annual sedimentation in our age model. As avalanche deposit is a very local phenomenon, the coring point has to be directly beneath the avalanche corridor to record the maximum number of events, thus capturing both drop stones and direct avalanche deposits. From our results, an avalanche deposit would be identified as multiple gravel elements at the same sediment depth, as opposed to a single element that could be related to a single rock falling from steep slopes. In order to better understand this deposition processes, multiple cores spatially dispersed in the deeper lake basin would give a better overall estimation.

When applying the age model to the LAU1104A sediment core, we are able to express the number of gravels identified since 1880 AD per 5-mm slice (Fig. 5). The gravel number goes from zero to almost twenty gravels

elements per 5 mm deposited in the lake floor. A total of 456 gravel elements were identified in the sediment core, 217 of which were identified outside flood layers. Yet, they represent a total of 106 922 mm$^3$ meaning 94.9% of the total measured volume. Gravels found in flood layers are thus characterized by a small size, probably related to lower competence transport mechanism, such as temporary tributaries on the steep slopes only active during a heavy precipitation event. We compared the evolution of gravel number in the annual sedimentation with historic records of winters with higher avalanche activity in the Oisans valley. The winter of 1922-1923 was an exceptional year in terms of winter precipitation in the Oisans valley, and avalanches destroyed numerous buildings and covered roads with thick snow deposits (Allix, 1923). The winter of 1969-1970 was also exceptional in terms of heavy snowfall, and no less than 800 avalanches were reported (Jail, 1970). On February 10$^{th}$, 1970, an avalanche killed 39 people, making it the most catastrophic avalanche in the last 200 years. In 1978, the Ecrins National Park rangers reported numerous avalanches in the Oisans valley, especially in spring with wet snow avalanches temporarily blocking roads (Ecrins national park internal report, 1978). The avalanche activity in the French Alps has also been explored based on the "Enquête Permanente sur les Avalanches" (EPA) since 1950. Four periods correspond to higher snow avalanche frequency in the northern French Alps: 1950-1955, 1968-1970, 1978-1988, and 1993-1998 (Eckert et al., 2013a) (Fig. 5). The nearest avalanche record is based on tree rings growth disturbance, located 10 km north from Lake Lauvitel which identified 20 events since 1919 AD (Corona et al., 2010) (Fig. 5). In Lake Lauvitel sediment sequence, the periods of increased numbers of rocks are around 1888, 1898, 1920-1931, 1939, 1949, 1970-1972, 1977-1980 and 1990-1993 AD (highlighted in blue). Considering our age model uncertainties (Fig. 4), these periods are in rather good agreement with higher avalanche activity from tree ring based calendar probably due to their proximity. Avalanches occur at a local scale (McCollister et al., 2003), but similarity between records was reported as far as 50 km distance (Butler and Malanson, 1985). In the meantime, the comparison with the EPA record seems more ambiguous. A recent study on tree ring based avalanches record tested the representatively of the natural archive to the meteorological conditions during the last fifty years based on the EPA data base (Schläppy et al., 2016). It revealed a underestimation compared to natural variability estimated to roughly 60% (Corona et al., 2012; Schläppy et al., 2014), and may be transferable to lacustrine avalanche deposits. Based on the comparison of our lake sediment record with other historical and tree rings calendar, the variability of avalanche occurrence in the Oisans valley seems to be represented by a minimum of four rocks present in a 5 mm thickness layer. It is thus quite difficult to reach solid conclusions regarding this relationship which is probably non-exhaustive, but may reflect part of the avalanche activity deposited in Lake Lauvitel. We thus need to develop both long-term and multiple site reconstructions of snow avalanche deposits to discuss its variability in terms of forcing mechanism. In these perspectives the CT scanning method appears to be a very promising tool.

The numerical counting method using the CT scan radiographs is well suited for this type of lacustrine sediment because density difference between fine silty and coarse gravel elements is quite significant. The resolution of the CT scan allowed identification of the centimeter-sized gravels found in sediment cores. Meanwhile, CT scan imagery was limited to a pixel resolution of only 500x500 μm due to the analysis of a volume as large a one-meter-long sediment core in our case. We highlight manual and numerical counting were in accordance on the absence of gravel-sized element in the sediment. Additionally, quantitative 3D imaging revealed useful to characterize gravel size elements related to instantaneous deposits but smaller ones are more difficult to discriminate as they are too

close to the pixel resolution. Some discrepancies between the manual and numerical gravel counting have to be noted in our study, which is probably related to the used image resolution. This constitutes one of the limits of this application based on the CT scan analysis, and better image resolution would then be more adapted to identify small gravels within the sediment core. This is the case for most of the OM macroremains identified in the sediment core

(Fig.3b-2), mainly composed of small roots or leaves characterized by an elongated and thin shape making them difficult to clearly identify with the used resolution of CT scan analysis. The largest OM elements located at the base of the thickest flood deposit were however clearly identified (Fig. 3b-1). As the analysis is based on relative density, some calibrations of known clastic or organic elements would be necessary in order to enhance qualitative information. In the end, the CT scan is a powerful non-destructive tool for investigating clastic elements in a

sedimentary core as well as OM rich levels. The clear potential in developing this method could be used in a wider range of quaternary sediments studies. Advantages of this method are for example the capacity to keep a numerical record of the sample before destructive analysis and the possibility to identify suitable depth for macro remains sampling for $^{14}$C analysis. This methodology opens new perspective for further natural archives studies to be used as complementary, effective tool to existing techniques.

**5 Conclusion**

CT scans are a well-developed analysis in the medical community and have been used for several geoscience-related studies in the past decades. The principle of the analysis is based on differences in the relative densities of an object. This study explores the possibility of using this technique on lake sediments to reconstruct instantaneous deposits. The analysis highlighted the presence of denser >2-mm mineralogical particles in the silty sedimentary matrix, as

well as the abundant organic matter which could be a useful tool for sampling macroremains for $^{14}$C analysis. The sedimentary analysis coupled with CT scanning of Lake Lauvitel sediment core, led to identify flood deposits, as well as the presence of poorly sorted layers accompanied with gravel size elements most likely related to wet snow avalanche deposits. However, the correspondence between historical and natural archives data presents some discrepancies. Exploration on both longer timescales and multiple site record would allow to better understand wet

snow avalanche past variability. The use of the CT scan methodology opens new perspectives in reconstructing past instantaneous deposits such wet snow avalanche in lacustrine sediments.

**6Acknowledgments**

L. Fouinat's PhD fellowship was supported by a grant from Ecrins National Park, Communauté des Communes de l'Oisans, Deux Alpes Loisirs and the Association Nationale de la Recherche et de la Technologie (ANRT). The

authors wish to thank Ecrins National Park for their authorization for sampling and assistance during the field work. The authors thank the Laboratoire Souterrain de Modane (LSM) facilities for the gamma spectrometry measurements and Hopitaux Universitaires de Genève (HUG) for the CT scan analysis.

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

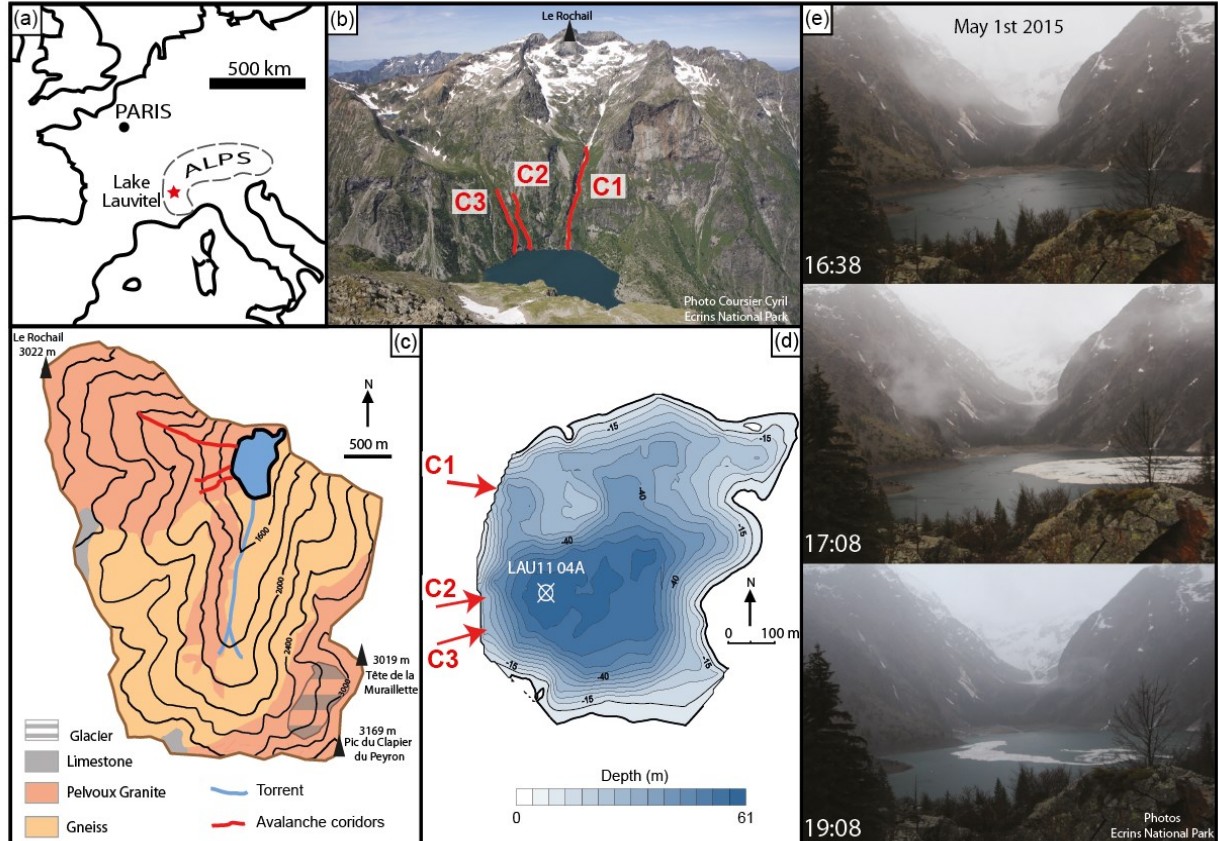

Figure 1: (a) Location of Lake Lauvitel, (b) Photo looking westward toward the location of the three avalanche corridors in the Lake Lauvitel watershed. (c) Simplified geologic map of the Lake Lauvitel watershed. (d) Lake Lauvitel bathymetric map and location of the three avalanche corridors and position of the LAU1104A coring point. (e) Photos of the lake looking to the south, with an avalanche entering the lake via the C1 corridor on May 1$^{st}$ 2015.

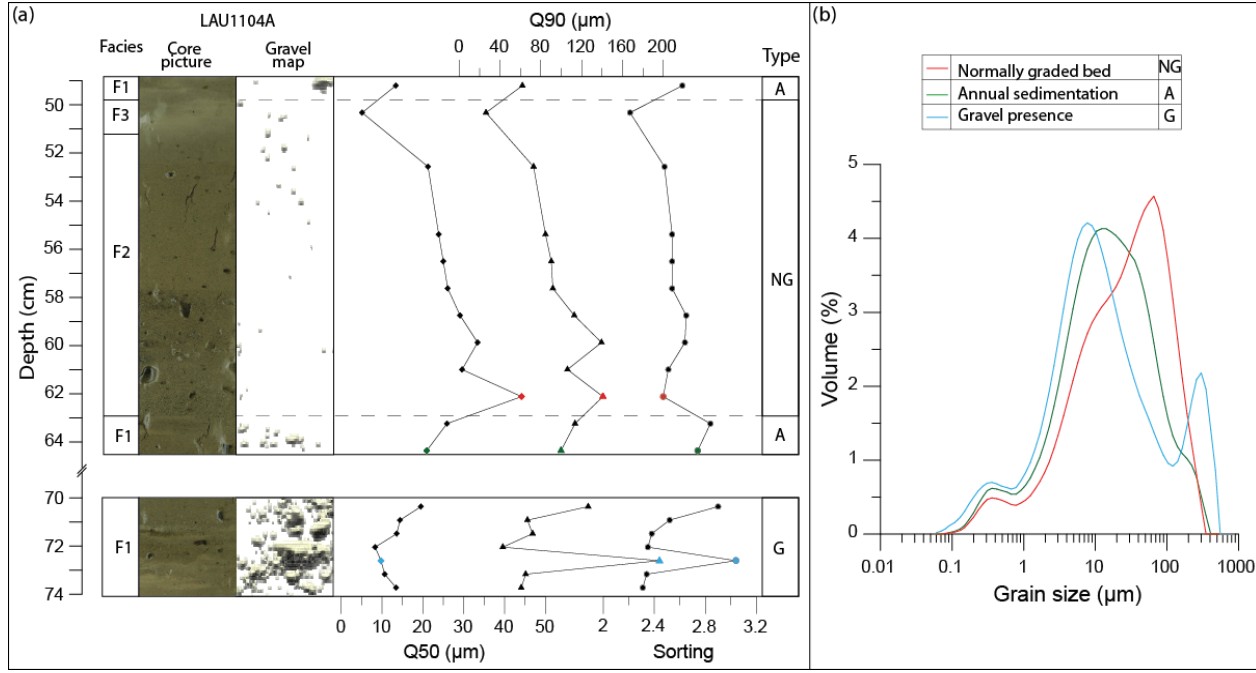

515

**Figure 2: (a) Characterization of typical facies of LAU1104A sediment core, based on Median grain size (Q50), $10^{th}$ percentile coarse grains (Q90) and sorting parameters. (b) Comparison between: NG-normally graded bed base sample (red line); A-annual sedimentation (green line) and G-gravel presence (blue line) grain size distributions**

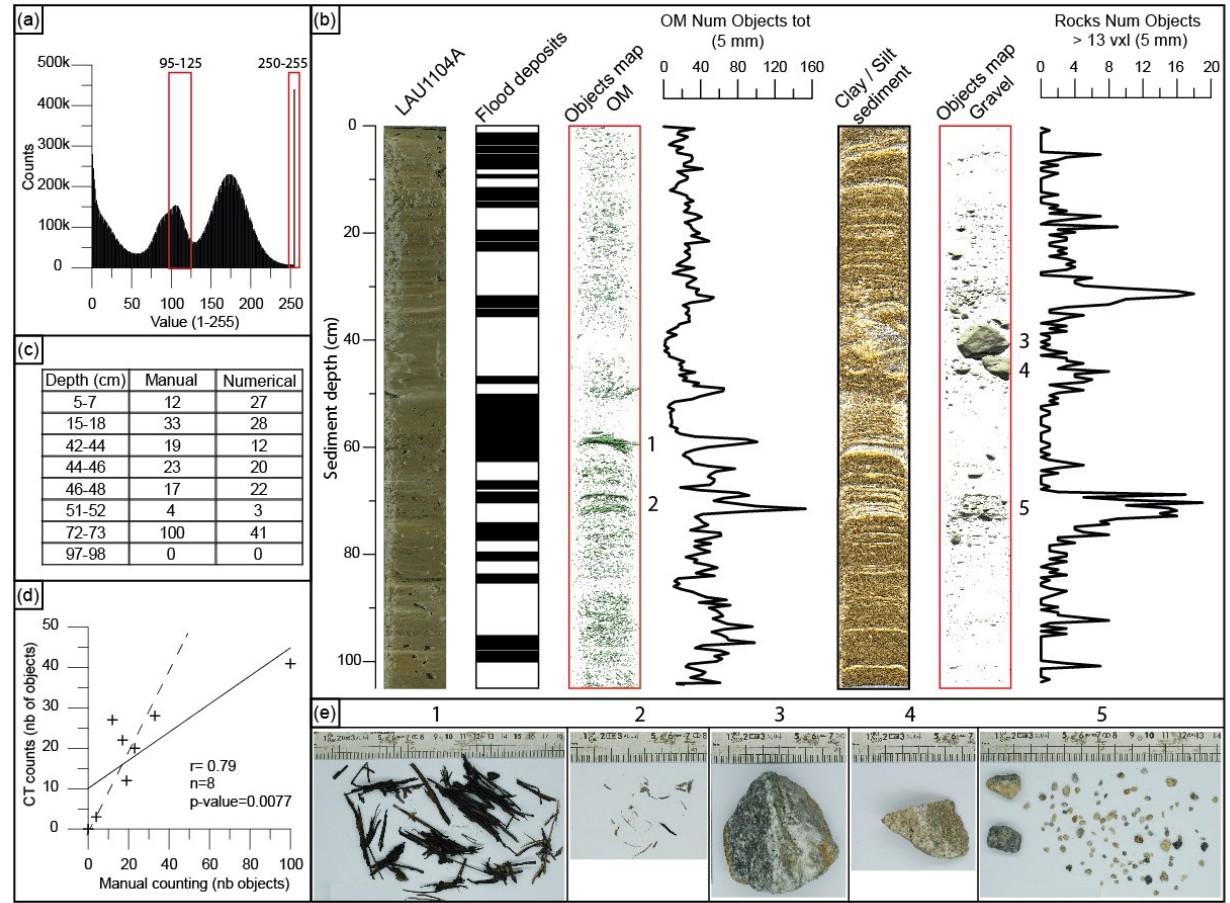

Figure 3: (a) Number of counts histogram for 1 to 255 levels of grey; selected range for OM (95-125) and for gravels (240-255) shown in red. (b) From left to right: core LAU1104A photography, position of flood deposits, CT image stacks of both rocks and OM and corresponding totals summed at 5 mm intervals. (c) Selected depth for comparison between manual and numerical counts in core LAU1104A. (d) Correlation between manual and numerical rock counts (solid line), CT counts = manual counts (dashed line) (e) Photographs of organic matter (e1, e2) and gravel-sized elements (e3, e4, e5) recovered from the LAU1104 sediment core.

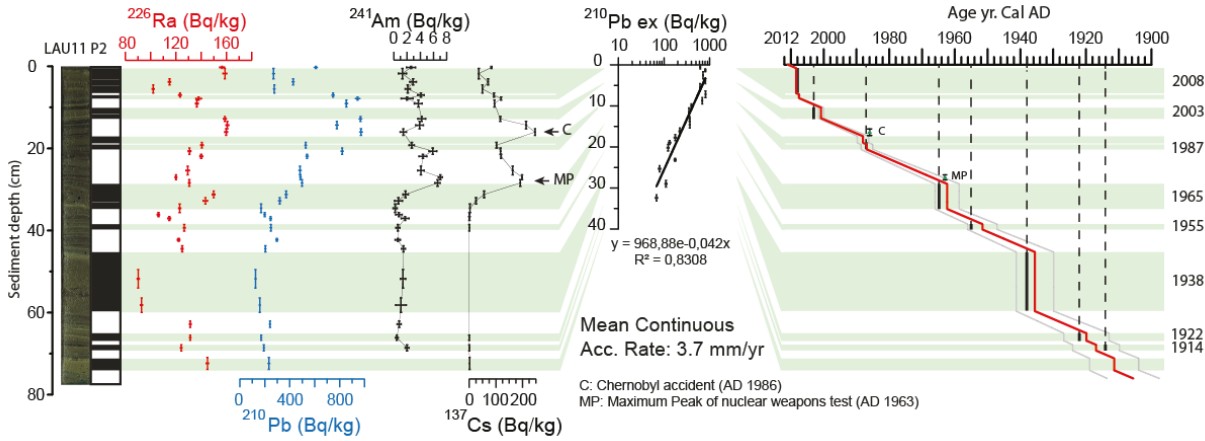

Figure 4: $^{226}$Ra, $^{210}$Pb, $^{241}$Am, and $^{137}$Cs activity profiles for core LAU11P2. Application of the CFCS model to the synthetic sedimentary profile of excess $^{210}$Pb (without normally graded beds, which are considered to be instantaneous deposits).

 **Resulting age-depth relationship with 1σ uncertainties and indications of historic flood dates associated with normally graded beds and the two artificial radionuclide markers.**

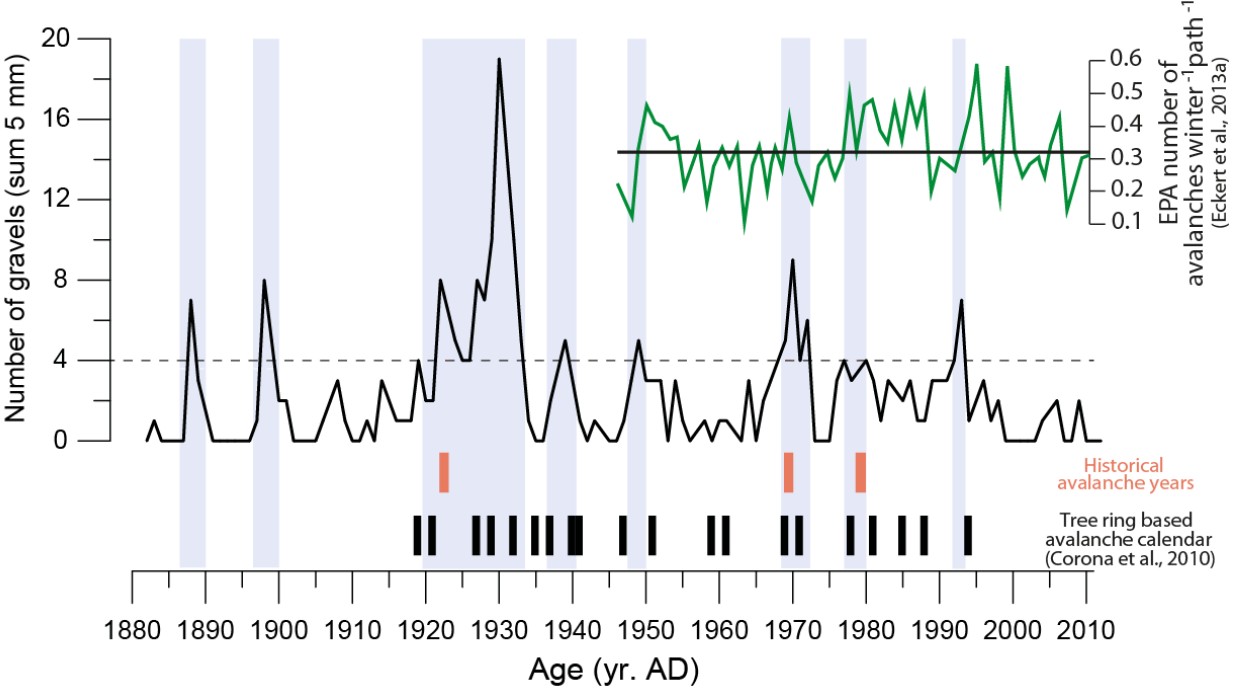

**Figure 5: Sum of gravels >13 voxels at 5 mm intervals identified in the LAU1104A sediment core since 1880 yr. AD without the normally graded beds. The dashed line represents the threshold number from which avalanche periods are identified (highlighted in blue). Exceptional winters found in the bibliography are represented in red (Allix, 1923; Jail, 1970; Ecrins National Park internal report, 1978). EPA number of avalanches per path since 1950 AD in green, interannual mean value in black (Eckert et al., 2013a). Avalanche record for the past century from tree rings in the nearby Romanche river valley (Corona et al., 2010).**