# Peer review of "A new CT scan methodology to characterize small aggregation gravel clast contained in soft sediment matrix"

_Earth Surface Dynamics, 2016_

## Referee Comment (RC1) · Anonymous Referee #1 · 24 Jun 2016

General Comments: The paper by Fouinat et al. explores a CT-based methodology for identifying snow avalanche deposits in lake sediments. To this end, the authors use an alpine lake with well-developed avalanche paths as their study test site. In general, the methodology is sound; however, as discussed below under "specific comments," this reviewer thinks the authors can improve their verification analyses and subsequently, the method's usefulness for other studies. This improvement includes the addition of detailed grain size measurements and measured organic matter (via LOI 550C or similar).

Unfortunately, the paper suffers also from very poor grammar, sentence structure, and organization. Without rewriting the paper myself – which is beyond the scope of this reviewer's expected work – the paper is not acceptable without a major edit to improve its readability.

[Figure]

Specific Comments (in order of paper not in order of importance or significance): 1. Introduction AND Background: this entire section requires significant development, referencing, and editing. For example, provide a better background on the dynamics of avalanches including the different types. What is a wet avalanche? 2. Line 76: the first sentence says "few remain known" and then you say there are "several studies"? requires editing. 3. Line 81 beginning with "These coarse particles...": requisite to this study is the clear demarcation of processes that generate coarse sediment deposits in lake sediments. Run-off, avalanches, debris flows, IRD...It is important to develop this section now, because later in the paper you discuss the presence of large gravel in flood deposits. Meanwhile, in the Line 81 section you seem to dismiss large gravel in flood deposits. Why the contradiction? Also, why is organic matter diagnostic of avalanche deposits? Did you measure organic matter via LOI 550C or equivalent? 4. Line 127: final sentence seems out of place? 5. Lines 149-154: Grain size! Where is the grain size data? Sampling interval? Show the data? it seems that grain size data would be required data to show and discuss for comparison to the "remote sensing" CT method. Verification is key to this methodologies usefulness. 6. Line 150: No grain size pretreatments!? Organics are a huge part of this core according to your discussion below...if so, they will introduce a false "grain size." Generally, pretreatments consist of organic removal, carbonate removal, and biogenic silica removal...BUT, at the very least, organic removal is expected. Explain and justify. SHOW the real, measured non-CT grain size data. 7. Line 151-154: Show the data. 8. Line 154.5: Organic matter is an important part of your discussion yet you have not actually measured organic matter? Use a simple analysis such as LOI 550C to determine the percent organic matter in your core for verifying the CT-scan estimate. Show the data. 9. Lines 159-161: Prove that these events are normally graded. As it reads, it seems these event's origin is arbitrarily defined. Show the grain size data. How did you define these events without any quantitative measurements? Are these layers simply based on visualization? Again, grain size, organic matter, >2mm grain size counts, etc...show the data. You require a completely new figure showing the core photograph versus it

physical data…show the CT data later OR in comparison to the physically measured data, such as grain size and organic matter. 10. Line 165: How were these facies defined? Can you better prove that thesis facies exist suing more than visual data? Can you plot these facies on one of your strat column figures. 11. Lines 168-170: Show us the proof for these normally graded turbidites. Grain size? 12. Lines 171-172: what are typical avalanche deposits? how are they not considered instantaneous like the flood event layers? 13. Line 173: "from the torrent"…what? 14. Line 174-175: Seems to contradict what you said earlier about gravel in flood deposits? so, multiple processes can co-occur? flood, debris flow? this makes the signal complex. How do you resolve this complexity with what you infer as "pure" avalanche deposits? 15. line 193: angular? The short transport paths support angular particles for any mass transport process. 16. Lines 216-218: nice verification method but why not more comparison depths? Key to this paper is proving that your "remote sensing" CT technique is an acceptable substitute for more time consuming physical sediment analyses. 17. Line 219: why no LOI 550C to verify these CT estimates??? 18. Line 264: seems like this could be more robust via a lot of more comparison depths? 19. Line 275: strangely worded? Do you mean gravel in between flood event layers or in the middle of a flood event layer?? 20. Lines 313-316: Seems redundant? 21. Figure 2: make sure the images match the figure captions.

Technical Corrections: Too many to correct. I leave it to the authors to seek an independent editor.

---

## Referee Comment (RC2) · Anonymous Referee #2 · 5 Jul 2016

Review of Fouinat et al., "A new methodology exploring the record of snow avalanches in lake sediments" for Earth Surface Dynamics

The manuscript of Fouinat et al., seeks to provide a new approach to reconstruct the past occurrence of snow avalanches from lake deposits. The introduction explains the challenges of documenting these events in many locations, and the importance of having historic context of their relative frequency. The paper argues that the CT-scan approach allows for a non-destructive and continuous (throughout a core) quantitative analysis of sedimentary records, in terms of grain size, number of grains, and assessing this for materials of different 'relative density'. This is quite interesting, namely in the potential to look at macro-organic debris (lower density) and clastic material separately, and isolate large clasts from the matrix in situ. Most of the time we try and get rid of the organic matter, and this study shows how it could provide interesting information. The data are well presented, the core chronology seems robust, and the data are interpreted alongside other records of avalanche occurrence. There are also links to the erosion of soil and vegetation (and organic matter/nutrients/carbon) which could interest a wider audience. The paper should be considered for Earth Surface Dynamics.

However, there are some issues which need to be addressed in my opinion:

1. The paper is set up as a 'new method', with the abstract and parts of the manuscript suggesting that the CT-scan method allows for 'avalanche deposit reconstruction'. But to do this, surely you need one (or more) known avalanche input events which have been cored to examine their sedimentology and make up of organic and clastic debris? (such as the one shown in May 2015 in Fig. 1?) I didn't see this being done clearly. Therefore, this paper does not provide a test of a method, but an interpretation of the sedimentology in terms of the process operating.

In my opinion, I think the paper would be better set up to illustrate how CT-scanning can be used to provide new, quantitative information on sedimentary deposits of a high altitude lake basin. This is in the context of interpreting fluvial events, debris flows, and avalanche deposits. I suggest refocusing the piece on the record itself, and interpreting it in terms of geomorphic and sedimentary processes. A paper with that focus would have to rely less on this being a transferable method just for 'snow avalanches', which is a weak part of the manuscript. Also, by doing so one might actually conclude that avalanches are pretty difficult to reconstruct in this deposit (probably because of frozen vs ice-free lake conditions when an event happens). A revised version could then focus more on better explaining and justifying the approach shown in Fig 2b, which seems to allow information on the distribution of organic matter (and its size) and certain clastic grain sizes.

2. More information needed on the CAT-scan methods and results: The '256 grey-scale' values are central to the approach here, providing a 'relative density'. There

needs to be more discussion of this – is this calibrated to anything? A standard of known density? Is it part of the 'soft tissue and bone kernal'. Why is the range 95-125 selected for organic matter – it seems to be justified on picking a single twig – what about finer organic debris, or leaves etc? There is a large second 'hump' in the histogram of values at 160 (Fig. 2a) – what is this value? How does the CAT-scan deal with a voxel which is made of air/water+silt – does this end up with a value <255, or lower? For this (and the subsequent counting methods) to be more widely applied, this needs more discussion/justification, and an awareness of caveats of this classification (i.e. quartz vs mica could have different density?).

3. The processes which deliver sediment/debris to the lake basin: The paper mentions that the lake is frozen for several months in the winter. Surely this makes it almost impossible to identify an avalanche deposit, because one entering an open lake will look fundamentally different to one which enters the lake after the thaw of the surface ice? The paper discusses this (at the start of the discussion), but to me this is the major caveat of the approach, and a flaw of focusing the paper solely on tracking avalanche deposits. It seems these can have a non-unique signature in the record.

4. Number of rocks as a proxy: This wasn't clearly justified as a proxy when introduced line 291. The link between the count data and the historical EPA dataset (which is regional??) shown on Fig. 4 is not convincing. I would use it to argue that it is extremely challenging to fingerprint these events at all in the record. The reason may well be that they do not have a unique sedimentary deposit associated with them. Why would >4 rocks reflect an avalanche and not a fluvial flood or debris flow? I'm not sure you can justify this.

5. The paper has numerous grammatical errors which need to be addressed.

Other comments:

52+54: snow vs wet avalanches – be clear on terminology and what it means.

56: 'Elements with different densities', perhaps 'assessment of the relative density of clasts in the deposit'

58: I wasn't convinced that the data showed that organic matter macro-remains characterise wet avalanches.

59: would perhaps be better to cast this in terms of new insight on sedimentary archives which can be used alongside existing approaches?

108: did you really test it?

117: do you mean using 10Be exposure ages – spell this out please.

146: what is the noise – this needs more explanation (as do other elements of this approach – see comment 2 above)

149: why is this grainsize data not shown?

180: after isolation – explain better what you mean here.

199: units are needed after density.

258+269: these sentences seem to contradict one another?

---

## Author Comment (AC1) · 2 Aug 2016

The comment was uploaded in the form of a supplement:

Please also note the supplement to this comment:
http://www.earth-surf-dynam-discuss.net/esurf-2016-25/esurf-2016-25-AC1-supplement.pdf

---

## Author Response (AR1)

*Author's responses to the referee comments are presented in this document. We would like to thank the referees for their remarks and we hope the revised manuscript has improved substantially. Referees comments are marked in black italic and author's responses are in blue.*

*Anonymous Referee #1

*General Comments: The paper by Fouinat et al. explores a CT-based methodology for identifying snow avalanche deposits in lake sediments. To this end, the authors use an alpine lake with well-developed avalanche paths as their study test site. In general, the methodology is sound; however, as discussed below under "specific comments," this reviewer thinks the authors can improve their verification analyses and subsequently, the method's usefulness for other studies. This improvement includes the addition of detailed grain size measurements and measured organic matter (via LOI 550C or similar).*

*Unfortunately, the paper suffers also from very poor grammar, sentence structure, and organization. Without rewriting the paper myself – which is beyond the scope of this reviewer's expected work – the paper is not acceptable without a major edit to improve its readability.*

We had the manuscript edited by AJE for the English prior to submission. We made changes in the grammatical errors found in the manuscript.

[Figure]

**EDITORIAL CERTIFICATE**

This document certifies that the manuscript listed below was edited for proper English language, grammar, punctuation, spelling, and overall style by one or more of the highly qualified native English speaking editors at American Journal Experts.

Manuscript title:
A new methodology exploring the record of snow avalanches in lake sediments

Authors:
Laurent Fouinat, Pierre Sabatier, Jérôme Poulenard, Jean-Louis Reyss, Xavier Montet, Fabien Arnaud

Date Issued:
March 31, 2016

Certificate Verification Key:
CA22-CA11-500D-418D-33B4

[Figure]

This certificate may be verified at www.aje.com/certificate. This document certifies that the manuscript listed above was edited for proper English language, grammar, punctuation, spelling, and overall style by one or more of the highly qualified native English speaking editors at American Journal Experts. Neither the research content nor the authors' intentions were altered in any way during the editing process. Documents receiving this certification should be English-ready for publication; however, the author has the ability to accept or reject our suggestions and changes. To verify the final AJE edited version, please visit our verification page. If you have any questions or concerns about this edited document, please contact American Journal Experts at support@aje.com.

American Journal Experts provides a range of editing, translation and manuscript services for researchers and publishers around the world. Our top-quality PhD editors are all native English speakers from America's top universities. Our editors come from nearly every research field and possess the highest qualifications to edit research manuscripts written by non-native English speakers. For more information about our company, services and partner discounts, please visit www.aje.com.

*Specific Comments (in order of paper not in order of importance or significance):*

*1. Introduction AND Background: this entire section requires significant development, referencing, and editing. For example, provide a better background on the dynamics of avalanches including the different types. What is a wet avalanche?*

**The introduction was modified, and we added supplementary information about wet avalanches. We hope this gives a better overlook of the wet avalanche phenomenon. The added text is:**

**"For most of wet avalanche, erosion occurs when the avalanche runs over bare ground or involves the whole thickness of the snow cover (Luckman, 1977).Two types are distinguished according to their water content, slush flows corresponding to a liquid mixture of mud and snow and wet avalanches described as non-water saturated snow flow moving as a solid mass composed of adjacent snow balls (Jomelli and Bertran, 2001). The mechanisms of erosion are various, abrasion, scratching and impact of the basal debris or even plucking unconsolidated rocks from the substratum; they can form distinctive geomorphic features such as narrow or funnel shaped gullies, debris covered slopes and small depression formed by repeated avalanche impacts called avalanche pits (Rapp, 1959; Luckman, 1977)."**

*2. Line 76: the first sentence says "few remain known" and then you say there are "several studies"? requires editing.*

**To improve readability we removed the sentence in the revised manuscript**

*3. Line 81 beginning with "These coarse particles. . .": requisite to this study is the clear demarcation of processes that generate coarse sediment deposits in lake sediments. Run-off, avalanches, debris flows, IRD. . .It is important to develop this section now, because later in the paper you discuss the presence of large gravel in flood deposits. Meanwhile, in the Line 81 section you seem to dismiss large gravel in flood deposits. Why the contradiction? Also, why is organic matter diagnostic of avalanche deposits? Did you measure organic matter via LOI 550C or equivalent?*

**We added text to the introduction paragraph to better explain the processes generating coarse sediment deposit in lake sediment.**
**Added text :**
 **"These coarse particles could have been incorporated into the sediment by different gravity transport processes e.g. i) related to liquid water such as high energy run off or debris flows competent enough to transport clay size to boulder size sediment ii) Rock fall transporting essentially sand to large boulders sediment but with low occurrence frequency iii) glaciers from clay to very large boulders iv) wet snow avalanches transporting clay size to gravel size sediment (Blair and McPherson, 1999; Vasskog et al., 2011). The incorporation of gravels transported to the lake will be different depending on the presence of ice. If lake is frozen the sediment would be deposited on top of the ice until thaw, then ice would spread around the lake and release clastic material falling to the lake bottom and creating drop stones (Luckman, 1975). If the lake is already freed from ice, the sediment would enter directly in the water creating a localized accumulation of sediment. In the case of debris flow, the resulted deposit was described as an underflow characterized by a normal gradation and well sorted sediments (Irmler et al., 2006). Another transport process non gravity related, would be particles trapped in ice on the lakeshore in the early winter, when ice melts in spring, the particles would be transported around the lake surface then fall at the lake bottom as drop stones."**

*4. Line 127: final sentence seems out of place?*

**This restricted area is set by the national park, environmental based rules seem important for the authors.**

*5. Lines 149-154: Grain size! Where is the grain size data? Sampling interval? Show the data? it seems that grain size data would be required data to show and discuss for comparison to the "remote sensing" CT method. Verification is key to this methodologies usefulness.*

We added the figure 2 showing the grain size data between 0-800 µm. Characterization of each facies is also shown in the figure. The grain size analysis of grain > 2 mm was done using a sieve. This is the grain size data comparable with the CT scan analysis only and comparison is shown on the Figure 3c and 3d.

6. Line 150: No grain size pretreatments!? Organics are a huge part of this core according to your discussion below...if so, they will introduce a false "grain size." Generally, pretreatments consist of organic removal, carbonate removal, and biogenic silica removal...BUT, at the very least, organic removal is expected. Explain and justify. SHOW the real, measured non-CT grain size data.
+
7. Line 151-154: Show the data.
+
8. Line 154.5: Organic matter is an important part of your discussion yet you have not actually measured organic matter? Use a simple analysis such as LOI 550C to determine the percent organic matter in your core for verifying the CT-scan estimate. Show the data.
+
17. Line 219: why no LOI 550C to verify these CT estimates???

We identified organic matter because it is part of the sediment, the aim of the manuscript was not to characterize the OM and make comparisons with the LOI 550C. OM in lake sediment is both composed of terrestrial OM and autochthon OM related to biological activity in the lake, this last part of OM is very small particles (less than 10µm) and not measured by CT scan, thus a direct comparison between LOI550 and CT estimation is not possible because a major part of OM is not quantifed by CT scan. In addition, the OM doesn't seem to be completely related to the avalanche deposits as we found also in flood deposits. This approach would be suitable for identifying organic macro remains for C14 dating for example. Most of the time we found small size twigs, roots or leaves, which are close or under CT-scan resolution. They are less suitable than coarse clastic sediment for characterization, this is why we didn't show LOI 550°C data. For information, LOI 550°C was measured on other sediment core from Lake Lauvitel and showed values from 3 to 8 %, with average value of 5 %. The OM concentration is thus very small in size, and it is very unlikely that the grain size analysis is biased by OM. But in the new version of the manuscript we show grain size data for typical flood and avalanche facies in a new figure see below

[Figure]

Figure 2: (a) Characterization of typical facies of LAU1104A sediment core, based on Median grain size (Q50), 10[th] percentile coarse grains (Q90) and sorting parameters. (b) Comparison between: NG-normally graded bed base sample (red line); A-annual sedimentation (green line) and G-gravel presence (blue line) grain size distributions

*9. Lines 159-161: Prove that these events are normally graded. As it reads, it seems these event's origin is arbitrarily defined. Show the grain size data. How did you define these events without any quantitative measurements? Are these layers simply based on visualization? Again, grain size, organic matter, >2mm grain size counts, etc. . .show the data. You require a completely new figure showing the core photograph versus it physical data. . .show the CT data later OR in comparison to the physically measured data, such as grain size and organic matter.*

*+*

*10. Line 165: How were these facies defined? Can you better prove that thesis facies exist suing more than visual data? Can you plot these facies on one of your strat column figures.*

*+*

*11. Lines 168-170: Show us the proof for these normally graded turbidites. Grain size?*

**Thank you for this comment; we added the figure 2 in the revised manuscript to show grain size data for typical flood, avalanche and annual sedimentation facies.**

**The evolution of the median grain size and the coarser 10$^{th}$ percentile along the graded bed allows us to consider them to be turbidities caused by heavy rainfall in the watershed (Støren et al., 2010; Giguet-Covex et al., 2012; Wilhelm et al., 2012b, 2012ba, 2013; Gilli et al., 2013; Wilhelm et al., 2015).**

*12. Lines 171- 172: what are typical avalanche deposits? how are they not considered instantaneous like the flood event layers?*

**Typical avalanche deposits were described on the ground e.g. Jomelli and Bertan (2001). After deposition in lake sediments it changes. We did not identify a specific facies on the sediment core, but the presence of coarse grain clast in the middle of the sediment core was revealed by the CT-scan analysis. We interpret the finely laminated sediment facies F1 (annual sedimentation). When rock fall into the lake sediment (related to avalanche) we can have two possibility 1/ the rock enter in the previously deposited sediment or 2/ as the avalanche occurring in spring, during summer and autumn sediment is deposit and filling the spaces between the rocks. It these both configurations we cannot consider sediment in between rock as instantaneous like flood sediment because of the annual sedimentation.**

*13. Line 173: "from the torrent". . .what?*

**We changed this to "Torrential activity" in the revised manuscript**

*14. Line 174- 175: Seems to contradict what you said earlier about gravel in flood deposits? so, multiple processes can co-occur? flood, debris flow? this makes the signal complex. How do you resolve this complexity with what you infer as "pure" avalanche deposits?*

**We added a paragraph to the discussion explaining the pure avalanche deposits and the difference with flood or debris flow deposits in the lake:**

**"The grain size analysis reveals a specific distribution concerning facies F2 representing the annual sedimentation. Sometimes gravel size sediment are present in the annual sedimentation, in this case the grain size distribution exhibits an additional mode in the sand class, almost no changes in the silt class and higher sorting values. The OM presence in the lake sediment is an addition of terrestrial and in situ origin, based on the CT-scan analysis there is no clear relation between OM and gravel layers except maybe in the thickest gravel accumulation (Fig. 3b-5). Jomelli and Bertran, (2001) observed the fine sediment associated to an avalanche is representing 6-16% of total sediment, which would explain why we have difficulty identifying them at the same depth as gravel elements. The largest OM element observed was located at the base of the thickest flood deposit (Fig. 3b-1), thus OM is not a distinctive parameter to identify an avalanche deposit. The grain size analysis in the same thick flood deposit revealed that the Q90 was still lower than coarsest grains in the gravel layers, despite the presence of few coarse elements in the upper part of the flood deposit**

(Fig.2a). Their presence may be explained by smaller intermittent tributaries activated by rainfall, such as in the avalanches corridors, transporting coarse grains to the lake bottom through a debris flow process or just by unconsolidated rocks transport on the steep slopes. Both floods and debris flows are characterized by the presence of a clay/silt sediment fraction and by normally graded lacustrine deposits well sorted (Gilli et al., 2013; Irmler et al., 2006) as opposed to a multi modal grain size distribution with high sorting values with no gradation as observed in lacustrine avalanches deposits in Norway (Vasskog et al., 2011). The deposits exhibiting those characteristics with the addition of gravel elements observed by the CT-scan analysis are interpreted as pure avalanche deposits for Lake Lauvitel. In order to identify only avalanches, we do not consider gravel elements present in the graded deposits."

15. line 193: angular? The short transport paths support angular particles for any mass transport process.

We agree with this statement. It is description of the rock found in the sediment core. It is precised angular because of short transport path supports the hypothesis that the origin of the rock is the nearby avalanche corridors.

16. Lines 216-218: nice verification method but why not more comparison depths? Key to this paper is proving that your "remote sensing" CT technique is an acceptable substitute for more time consuming physical sediment analyses.
+
18. Line 264: seems like this could be more robust via a lot of more comparison depths?

We are agree with the reviewer comment this manual counting is a good verification of the method, this manual counting comparison is time consuming and we prefer focus on depth with different range of gravel in spite of counting all depth. This why choose to apply this verification on 8 sediment depths presenting different range of gravel, between no gravel and the depth where the maximum gravels where identified by CT scan analysis (Fig.3c).

19. Line 275: strangely worded? Do you mean gravel in between flood event layers or in the middle of a flood event layer??

We removed "right" in the sentence of the revised manuscript

20. Lines 313-316: Seems redundant?

Changes were made in the revised manuscript

21. Figure 2: make sure the images match the figure captions. Technical Corrections: Too many to correct. I leave it to the authors to seek an independent editor.

We changed the captions in order to match the images.
*Review of Fouinat et al., "A new methodology exploring the record of snow avalanches in lake sediments" for Earth Surface Dynamics*

*The manuscript of Fouinat et al., seeks to provide a new approach to reconstruct the past occurrence of snow avalanches from lake deposits. The introduction explains the challenges of documenting these events in many locations, and the importance of having historic context of their relative frequency. The paper argues that the CT-scan approach allows for a non-destructive and continuous (throughout a core) quantitative analysis of sedimentary records, in terms of grain size, number of grains, and assessing this for materials of different 'relative density'. This is quite interesting, namely in the potential to look at macro-organic debris (lower density) and clastic material separately, and isolate large clasts from the matrix in situ. Most of the time we try and get rid of the organic matter, and this study shows how it could provide interesting information. The data are well presented, the core chronology seems robust, and the data are interpreted alongside other records of avalanche occurrence. There are also links to the erosion of soil and vegetation (and organic matter/nutrients/carbon) which could interest a wider audience. The paper should be considered for Earth Surface Dynamics.*

*However, there are some issues which need to be addressed in my opinion:*

*1. The paper is set up as a 'new method', with the abstract and parts of the manuscript suggesting that the CT-scan method allows for 'avalanche deposit reconstruction'. But to do this, surely you need one (or more) known avalanche input events which have been cored to examine their sedimentology and make up of organic and clastic debris? (such as the one shown in May 2015 in Fig. 1?) I didn't see this being done clearly. Therefore, this paper does not provide a test of a method, but an interpretation of the sedimentology in terms of the process operating. In my opinion, I think the paper would be better set up to illustrate how CT-scanning can be used to provide new, quantitative information on sedimentary deposits of a high altitude lake basin. This is in the context of interpreting fluvial events, debris flows, and avalanche deposits. I suggest refocusing the piece on the record itself, and interpreting it in terms of geomorphic and sedimentary processes. A paper with that focus would have to rely less on this being a transferable method just for 'snow avalanches', which is a weak part of the manuscript. Also, by doing so one might actually conclude that avalanches are pretty difficult to reconstruct in this deposit (probably because of frozen vs ice-free lake conditions when an event happens). A revised version could then focus more on better explaining and justifying the approach shown in Fig 2b, which seems to allow information on the distribution of organic matter (and its size) and certain clastic grain sizes.*

Sediment core were sampled after the avalanche event of May 1$^{st}$ 2015, but unfortunately no rocks were identified at the sediment-water interface. The avalanche was originating from the C1 corridor, which is located in the northern part of the lake where an upper basin is present. It is most likely that there is no sedimentary connexion between the upper basin and the lower basin (coring site). Thus this avalanche do not present any influence on sediment in the deeper basin, this is why we do not find any gravel in a sediment core taken after this avalanche in the deeper basin. The mechanism of gravel transport to the lake is demonstrated as avalanches, because of the presence of avalanche corridors, direct observations from National Park rangers, and the grain size results. Rocks brought by floods or in debris flow would be also accompanied by fine sediment clast, and this is not the case. The figure 2 exhibits the grain size distribution in the 0-800µm range; the addition of a sand class fraction is demonstrated.

The approach on Fig.2 is the base analysis for identifying coarse grains in the sediment cores. The authors wish to explore the mechanisms related to this gravel income in the Lake lauvitel in order to make this gravel fraction useful for further studies on lake sediments. As it is a first publication with this approach some questions remain unanswered, but we hope the revised manuscript will have better argumentation on the avalanche identification.

*2. More information needed on the CAT-scan methods and results: The '256 greyscale' values are central to the approach here, providing a 'relative density'. There needs to be more discussion of this – is this calibrated to anything? A standard of known density? Is it part of the 'soft tissue and bone kernal'. Why is the range 95- 125 selected for organic matter – it seems to be justified on picking a single twig – what about finer organic debris, or leaves etc? There is a large second 'hump' in the histogram of values at 160 (Fig. 2a) – what is this value? How does the CAT-scan deal with a voxel which is made of air/water+silt – does this end up with a value <255, or lower? For this (and the subsequent counting methods) to be more widely applied, this needs more discussion/justification, and an awareness of caveats of this classification (i.e. quartz vs mica could have different density?).*

**The soft tissue and bone kernel is a computer analysis made by the CT-scan on raw data which enhance the contrast and makes a better representation of both soft tissues and bones in a human body. We added the reference (Tins, 2010) which explains the technical aspects of CT-imaging. The use of this kernel represented the best image and identification of the clasts in the sediment core. We did not have time to develop further tests on the CT-scan because we were restricted to a limited time in the hospital radiology department, especially calibrating the signals with some known objects such as water, sand, air bubbles and identified organic matter elements. Those tests would certainly be very useful in the near future to better characterise the elements in the sediment core with precise density estimation if it is possible.**
**From what we have measured the sediment core analyzed in the CT-scan, the air around the sediment core appeared as black in the resulted images, meaning density 0.**

**The range 95-125 is selected by identification of macroremains: pinus twig several centimeters long and smaller roots and leaves recovered in the sieve during manual counting.**

**The hump at 174 is already explained in the text (line 182-184) "The second mode, centered on the 174 value, is relatively denser than OM. Its large spectrum and high count values correspond to the most common element in the sediment core, which would be the silty clay sedimentation matrix (Fig. 2b)."**

**Concerning the caveats we agree with the comments, some calibration is necessary in order to explore more precisely the densities. In this manuscript this is not our objective, we aimed at demonstrated that this tool is suitable for reconstruction gravel clasts elements deposition in the lake Lauvitel. We added in the discussion part an awareness of caveats:**

**"This is the case for organic matter, which was identified in the sediment core (Fig.2b-2), mainly composed of small roots or leaves characterized by an elongated and thin shape, making them difficult to clearly identify with the CT-scan analysis. (…)For further analysis of sedimentary elements, some calibration of known elements would be necessary. The density differences between different rock types or different organic macro remains doesn't allow further interpretation. "**

*3. The processes which deliver sediment/debris to the lake basin: The paper mentions that the lake is frozen for several months in the winter. Surely this makes it almost impossible to identify an avalanche deposit, because one entering an open lake will look fundamentally different to one which enters the lake after the thaw of the surface ice? The paper discusses this (at the start of the discussion), but to me this is the major caveat of the approach, and a flaw of focusing the paper solely on tracking avalanche deposits. It seems these can have a non-unique signature in the record.*

**Thank you for those comments; we tried to give a better explanation on the processes involved in the transport of gravels into the lake. The possible mechanisms are numerous (flood, debris flow, IRD or simply unconsolidated rocks fall), with the addition of the new figure 2 and the revised discussion paragraphs, we expressed different possibilities.**

**"The grain size analysis in the same thick flood deposit revealed that the Q90 was still lower than coarsest grains in the gravel layers, despite the presence of few coarse elements in the upper part of the flood deposit (Fig.2a). Their presence may be explained by smaller intermittent tributaries activated by rainfall, such as in**

the avalanches corridors, transporting coarse grains to the lake bottom through a debris flow process or just by unconsolidated rocks transport on the steep slopes. Both floods and debris flows are characterized by the presence of a clay/silt sediment fraction and by normally graded lacustrine deposits well sorted (Gilli et al., 2013; Irmler et al., 2006) as opposed to a multi modal grain size distribution with high sorting values with no gradation as observed in lacustrine avalanches deposits in Norway (Vasskog et al., 2011). The deposits exhibiting those characteristics with the addition of gravel elements observed by the CT-scan analysis are interpreted as pure avalanche deposits for Lake Lauvitel. In order to identify only avalanches, we do not consider gravel elements present in the graded deposits. The organic matter is usually associated with the fine sediment in an avalanche deposit, but the CT-scan analysis on relative lower densities attributed to organic matter are not always coinciding with avalanche deposits. Only the thickest gravel accumulation shows clear increase of organic matter content as well as increase in gravel number. The organic matter income is also related to torrential activity, as observed in the (Fig. 3b-1) where a pinus twig was found. Fine lamination (F1) is present in the split sediment core despite of the presence of gravel-sized elements, which is remarkable considering the volume of the gravel class and the deposition on the lake bottom. Two possible explanations would be i) integration of gravel elements without any sediment disturbance or ii) the spring avalanche would be deposited on the top of the sediment-water interface, then the summer and autumn finely laminated sediment would be deposited posterior to the gravel and sand clasts and filling the spaces between gravels. In any case, the basal deposit level would then represent the date of the avalanche deposit.

Wet avalanches occur at spring season when the snow pack is becoming unstable due loss of cohesion in the structure caused by warmer temperatures. It is also thaw season for lake ice. Hence, the avalanches could either be deposited on ice or enter directly in the water, as observed during the May 1st 2015 avalanche. At that time, the snow flow originated from the C1corridor in the northern part of the lake where an upper basin is present. It is likely to have no sedimentary connection with the deeper basin where the coring point is located. If the avalanche was deposited on lake ice, sediment may have been dispersed in the deeper basin as ice rafted debris and thus be recorded in a sedimentary core. Snow avalanche materials can be integrated into lacustrine sediments in two ways. In the case of a frozen lake, surface avalanche deposits are spread across the ice and subsequently drop to the lake sediment from drifting ice. When an avalanche occurs while the lake is ice-free, the avalanche deposits directly enter the water, and particles are concentrated in a more restricted area closer to the avalanche corridor. As this is a very local phenomenon, the coring point has to be directly beneath the avalanche corridor to record the maximum number of events, thus capturing both drop stones and direct avalanche deposits. An avalanche deposit would be identified as multiple gravel elements at the same sediment depth, as opposed to a single element that could be related to a single rock falling from steep slopes. In order to comprehend this deposition processes, multiple cores spatially dispersed in the deeper lake basin would give a better overall estimation. "

*4. Number of rocks as a proxy: This wasn't clearly justified as a proxy when introduced line 291. The link between the count data and the historical EPA dataset (which is regional??) shown on Fig. 4 is not convincing. I would use it to argue that it is extremely challenging to fingerprint these events at all in the record. The reason may well be that they do not have a unique sedimentary deposit associated with them. Why would > 4 rocks reflect an avalanche and not a fluvial flood or debris flow? I'm not sure you can justify this.*

Both the debris flow and the flood have a good sorting and graded deposit, as we identified normally graded beds we exclude them from the sediment record in order to keep only gravels in the annual sedimentation. We argue in the new discussion paragraph about the number of gravel to identify avalanche deposits. We added additional avalanche deposits based on tree ring record few kilometers from the study site (Corona et al., 2010) in order to have a better comparison. We agree that the main problem of the avalanche deposits in Lake Lauvitel is the different deposits, but the number of rocks present in the sediment core (499) is quite important and makes a significant part of the sedimentation. After comparison with other records, the number of 4 rocks in a 5 mm slice is coinciding with historical data, tree ring record and regional EPA. In Lake Lauvitel surely all avalanches are not recorded, the coring point situated in the deeper part of the lake doesn't allow identification of each avalanches. Avalanche records in bibliography are rare also expressing the difficulty to identify them no matter the methodology. Our method suggests a fast, non destructive way to identify them in lake sediments. As It is a new approach and a first publication some issues need to be addressed in the text and we added them.

" The number goes from zero to almost twenty gravel elements per 5 mm deposited in the lake floor. Possible avalanche deposits would be related to various gravel accumulation as the deposition of avalanches can occur on lake ice or directly in the water. However, a total of 499 gravel elements were identified in the sediment core, income is thus related to a frequent mechanism transporting them to the lake. (…)This gravel limit number seems to correspond at least to major avalanche events recorded in the valley, despite the variability in the deposition (ice or water), the various exposures of avalanche corridors, and various analytical measurements. We thus set four rocks as the minimum for recording wet avalanche deposits in Lake Lauvitel as a non-exhaustive identification, constituting first avalanche record in lake sediments for the French Alps."

*5. The paper has numerous grammatical errors which need to be addressed.*

This manuscript was edited for the English prior to submission by AJE, the certificate is added to the file. We made changes on the grammatical and typing errors found in the manuscript.

*Other comments: 52+54: snow vs wet avalanches – be clear on terminology and what it means.*

The introduction was modified, and we added a part concerning terminology of wet avalanche and definitions:

"Avalanche erosion only occurs when the avalanche runs over bare ground or involves the whole thickness of the snow cover (Luckman, 1977), this is the case for most wet avalanches. Two types are distinguished according to their water content, slush flows corresponding to a liquid mixture of mud and snow and wet avalanches described as non-water saturated snow flow moving as a solid mass composed of adjacent snow balls (Jomelli and Bertran, 2001). The mechanisms of erosion are various, abrasion, scratching and impact of the basal debris or even plucking unconsolidated rocks from the substratum; they can form distinctive geomorphic features such as narrow or funnel shaped gullies, debris covered slopes and small depression formed by repeated avalanche impacts called avalanche pits (Rapp, 1959; Luckman, 1977)."

*56: 'Elements with different densities', perhaps 'assessment of the relative density of clasts in the deposit'*

Thank you for the comment  changes were made in the revised manuscript

58: I wasn't convinced that the data showed that organic matter macro-remains characterise wet avalanches.

Many references are describing this organic matter in avalanches. But we agree with the reviewer, in Lake Lauvitel case, it is not so clear. We identified organic macroremains, but not only originating from avalanche deposits, but also brought to the lake bottom by floods or runoff. Thus, we do not use the presence of organic matter as an additional proxy characterizing avalanche deposits in the Lake Lauvitel. We removed macro remains from the abstract.

*59: would perhaps be better to cast this in terms of new insight on sedimentary archives which can be used alongside existing approaches?*

We agree, changes were made in the revised manuscritpt

*108: did you really test it?*

We have made CT-scan analysis on the Lauvitel sediment core because we thought that the number of rocks and gravel was important. We were surprised to identify so many gravel elements in the lake sediment

which is usually clay/silt/sand size. This was a first test for us. Further tests would be required in order to better characterize each elements in the sediment core, such as calibration on known density rocks, organic matter and water or air bubbles for example. We added text in the discussion paragraphs to express that:

"For further analysis of sedimentary elements, some calibration of known elements would be necessary. The density differences between different rock types or different organic macro remains do not allow further interpretation."

*117: do you mean using 10Be exposure ages – spell this out please.*

Yes, we mean 10Be exposure ages. Changes were made on the revised manuscript.
"4.7±0.4 kyr [10]Be exposure age (Delunel et al., 2010)"

*146: what is the noise – this needs more explanation (as do other elements of this approach – see comment 2 above)*

The noise is related to the analysis; some isolated voxels are presenting different densities compared to the 26 neighbour voxels. This is why we apply the despeckle filter in order to make the signal smoother. The plugin details are already expressed in the references (Bolte and Cordelieres, 2006; Tins, 2010). In the manuscript we wish to express how we used the plugin in order to identify the different density clasts in the sediment core.

*149: why is this grainsize data not shown?*

**We added the Figure 2 to present the grain size data.**

[Figure]

Figure 2: (a) Characterization of typical facies of LAU1104A sediment core, based on Median grain size (Q50), 10[th] percentile coarse grains (Q90) and sorting parameters. (b) Comparison between: NG-normally graded bed base sample (red line); A-annual sedimentation (green line) and G-gravel presence (blue line) grain size distributions

Result section:  we added the Figure 2 that is representing the grain size data of each facies identified. The following text was added:

The core lithology is composed of three facies (Fig. 2a). Facies 1 (F1) is silty-clay, dark-brown, finely laminated layer. It is interbedded by two other facies that are almost always associated with each other: Facies 2 (F2) is a normally graded bed from coarse sand to silt, sometimes with an erosive base; this facies is always associated with a thin white clay-rich layer Facies 3 (F3) on the top. Fig. 2b presents typical normally graded beds with grain size distribution (in red) characterized by a median grain size (Q50) of 44.1 μm and a mode of 81 μm. F1 (in green) exhibits a median grain size of 13.5 μm and a mode of 11.9 μm. Sometimes, F1 presence coincides with coarse gravel in the sediment, then the median grain size is similar 9.7 μm, but two modes are discernible at 7.2 and 258 μm. Sorting reveals different values depending on the deposit type; 2.50 average in the normally graded beds, 2.65 for the annual sedimentation and 3.05 for annual sedimentation with gravel presence.  The small difference in median grain size between annual sedimentation with and without gravel suppose limited addition in the fine grains fractions, but fraction superior to 100 μm and bad sorting and Q90  reveals a significant addition of sand size grains in the gravel layers.

 *180: after isolation – explain better what you mean here.*

We added "After selecting this mode, we isolated the numerical values in order to map them by using the plugin". The maps shown on figure 3b, are realized from the 95-125, the hump at 174 and 240-255 histogram values were isolated from the rest of the core. This process reveals the location of each pixel of this value in the numerical sediment core.

*199: units are needed after density.*

We changed the density by Volumetric mass density and added units: $t/m^3$

*258+269: these sentences seem to contradict one another?*

We made the text clearer about the use of organic matter in order to identify the avalanche deposits, in the end it is not a reliable enough proxy to clearly identify them along with gravel presence.

Discussion "The OM presence in the lake sediment is an addition of terrestrial and in situ origin, based on the CT-scan analysis there is no clear relation between OM and gravel layers except maybe in the thickest gravel accumulation (Fig. 3b-5). Jomelli and Bertran, (2001) observed the fine sediment associated to an avalanche is representing 6-16% of total sediment, which would explain why we have difficulty identifying them at the same depth as gravel elements. The largest OM element observed was located at the base of the thickest flood deposit (Fig. 3b-1), thus OM is not a distinctive parameter to identify an avalanche deposit."

 "This is the case for organic matter, which was identified in the sediment core (Fig.2b-2), mainly composed of small roots or leaves characterized by an elongated and thin shape, making them difficult to clearly identify with the CT-scan analysis at least with the used resolution."

---

## Author Response (AR2)

**Associate Editor Decision: Reconsider after major revisions (19 Aug 2016) by Valier Galy**
Comments to the Author:

Dear Dr. Fouinat,
I have now had a chance to carefully read your response to the reviewers' comments as well as your revised manuscript. I concur with both reviewer that – even after substantial improvements during the revision process - the general organization and focus of the manuscript is not optimal. While the data and in depth analysis presented are of clear interest with respect to understanding lacustrine sediment records and the erosion/deposition processes that generate them, it is not clear to me that the manuscript convincingly lays the ground work for using the proposed technique to reconstruct avalanche activity. This in part due to a lack of direct characterization of deposits that could be unambiguously attributed to avalanches. In the end, the comparison of independent proxies of avalanche activity with your CT scan based record (i.e. figure 5) is rather unconvincing. Therefore, in order for the paper to become suitable for publication in Earth Surface Dynamics, I suggest that you consider a significant re-organization of the manuscript and in particular a broadening of the focus of the study, along the lines of what Reviewer #2 suggested.

Sincerely,
Dr. Valier Galy

Dear associate editor Dr. Galy,

My co-authors and I wish to thank you for useful comments on the previous manuscript which were applied in our revised manuscript after the first answer to reviewer comments. We have made major modifications to the manuscript previously submitted to Esurf journal "A new methodology exploring the records of snow avalanches in lake sediments". We changed the scope of the article based on comment 1 from Reviewer #2 and editor's advice. The previous version was focusing on a new methodology in order to identify wet avalanches deposits in high elevation lake sediments. The revised manuscript is now oriented in order to give an interpretation of the sedimentology in terms of the process operating rather than the test of the method. The title is now changed to "A new CT scan methodology to characterize small aggregation gravel clast contained in soft sediment matrix". The introduction and discussion paragraphs were modified according to previous explanation. The introduction is now set up as a presentation of the CT scan method and previous applications in sedimentary studies. We then propose this method to better characterize mass-wasting events characterized by coarse clastic components due to their high transport capacity in lacustrine deposits. New literature was added accordingly. We cite three different transport mechanisms: floods, debris flow and wet avalanches and related underwater lacustrine deposits found in the literature. The revised discussion paragraph is now exposing the event related deposits recovered from our sedimentary record. Based on our results, we identified two different deposits, flood layers and gravel accumulation associated with poorly sorted finer grain size in the annual sedimentation. Based on literature, the latter seems to be related to avalanches deposits. We make comparisons with historical and natural archives (tree rings) avalanches records presented in Fig. 5 and seems in a rather good agreement between historical and tree ring based calendars. This leads to a discussion paragraph on the comparison with other records. The last part of the discussion is based on our CT scan method which is in the end suitable for identification and quantification of organic matter and different grain size clastic sediment. We emphasize the applied methodology opens new perspectives for further natural archives studies as complementary effective tool to existing techniques. My co-authors and I hope the changes in the scope of the manuscript are making the revised version stronger and more appealing to a broader audience. We also wish to thank you to still consider the article as a publication in your journal.

Sincerely,
Laurent Fouinat

*Anonymous Referee #2

*1. The paper is set up as a 'new method', with the abstract and parts of the manuscript suggesting that the CT-scan method allows for 'avalanche deposit reconstruction'. But to do this, surely you need one (or more) known avalanche input events which have been cored to examine their sedimentology and make up of organic and clastic debris? (such as the one shown in May 2015 in Fig. 1?) I didn't see this being done clearly. Therefore, this paper does not provide a test of a method, but an interpretation of the sedimentology in terms of the process operating. In my opinion, I think the paper would be better set up to illustrate how CT-scanning can be used to provide new, quantitative information on sedimentary deposits of a high altitude lake basin. This is in the context of interpreting fluvial events, debris flows, and avalanche deposits. I suggest refocusing the piece on the record itself, and interpreting it in terms of geomorphic and sedimentary processes. A paper with that focus would have to rely less on this being a transferable method just for 'snow avalanches', which is a weak part of the manuscript. Also, by doing so one might actually conclude that avalanches are pretty difficult to reconstruct in this deposit (probably because of frozen vs ice-free lake conditions when an event happens). A revised version could then focus more on better explaining and justifying the approach shown in Fig 2b, which seems to allow information on the distribution of organic matter (and its size) and certain clastic grain sizes.*

We agree the study lacks a direct observation of an avalanche and related deposit sampling. We have taken sediment cores after the avalanche event on May 1st 2015, but unfortunately no gravels were identified in the samples. As explained in the revised manuscript (l.130-131), the observed avalanche originated from the C1 corridor directed to the upper lacustrine basin where we were not able to sample because of large boulders and absence of fine sediment). The sedimentrary connection is supposed to be null between upper and deeper basin. We agree on the comment that the article does not provide a test of the method based on recognition of specific avalanches deposits. Based on this comment we oriented our subject on the method based on the CT scan to provide quantitative information on high energy sedimentary deposits of high elevation lakes and we change the title. The revised version is now focusing on the identification of flood deposits and mass-wasting events based on the CT scan results. The main changes are the introduction and the discussion paragraphs. The introduction is now exposing the CT scan as a new relevant method through examples of recent studies exploring quaternary sediment structure and quantifying components. The second part of the introduction focuses on the high energy lacustrine deposits likely to transport and deposits coarse grains. These are cited as floods (l.83), debris flow (l.92) and wet avalanches (l.97). The objectives of the study, exposed at the end of the paragraph, are to use of the CT scan methodology as a fast, nondestructive method to obtain quantitative information about sedimentary components, and especially on coarse grains. The application of this complementary to existing methods is applied on the Lake Lauvitel sediment. The watershed presenting steep slopes and specific forms of high energy events seems suitable for the study.

The discussion paragraph was also changed in order to interpret the observed deposits and deduce the mechanism related to those deposits. In the first paragraph (l.262-275), we identify turbidites deposits to link them to flood deposits through grain size analysis. The CT scan method observed some gravels in those flood deposits, and we discuss their origin which is excluded to be related to debris flow based on sediment accumulation. On the second paragraph (l.276-294) we identified sediment layers characterized by fine sediment poorly sorted exhibiting multi modal grain size distribution as well as numerous gravels. This type of deposits was described in the literature as avalanches deposits. We then discuss the different deposition mechanisms which may be involved. The third paragraph (l.296-328) is based on the Fig. 5, presenting historical and natural archives avalanches records found in literature compared to our gravel layers in Lake Lauvitel. The nearest avalanche record is from Corona et al., (2010), and exhibits a precise chronology. The direct comparison seems rather good with tree ring based calendar but ambiguous with the EPA record. This last feature was cited in the natural archives of tree rings which underestimates the total avalanches events by around 60%. In the discussion, the minimum gravel number to identify is based on the comparison with other records. The fourth paragraph (l.329-349) is presenting the limits of the CT scan method in our study, which is the image resolution for small elements identification and the satisfactory but perfectible relationship between manual and numerical counting. The advantages are also cited as identification of different sediment components such as gravel and organic matter in a quantitative method. The end of the discussion focuses on the perspectives of the method itself and growing potential for further studies on quaternary sediments.

---

## Author Response (AR3)

Dear Dr. Galy,

Thank you very much for your previous letter and the publication decision for the article. My co-authors and I have made the few technical corrections you recommended in order to publish the article. I would like to apologize for the delay for sending back the corrections. The re-read from an English native took a little more time than expected, due to a long travel from France back to Australia. The changes made in the manuscript are detailed in green in the revised manuscript. My co-authors and I hope those changes will make the manuscript more suitable for publication. Answers to the associate editor's comments appear in bold.

Sincerely,
Laurent Fouinat

**Associate Editor Decision: Publish subject to technical corrections** (07 Feb 2017)
by Valier Galy

Comments to the Author:

Dear Dr. Fouinat,

thank you for submitting a revised version of your manuscript. I have now had a chance to read it carefully and I am pleased to recommend its publication subject to technical corrections. Overall you did a very good job at broadening the scope the manuscript and I am sure it will make it much more appealing to the broad readership of Earth Surface Dynamics. I am aware that neither you nor your co-authors are native English speakers and – being myself in that position - I know how difficult it is to eliminate syntax errors. That said, before the manuscript can be published, please have it checked and edited by a native English speaker, in particular the sections that were largely rewritten during the revision process. A handful of minor comments also follow for your consideration.

L72: please define voxel as the average reader of Earth Surface Dynamics probably not familiar with this term.

**We added in the revised text (l.72) "(i.e. volumetric pixel)" after the word voxel in order to define the term used in the study.**

L209-210: it would be easy to accurately measure the actual volume, thereby removing the assumption on its density.

**We agree with your comment. The simpler and more accurate way to calculate the volume of the gravel piece, is to measure the weight of water displaced when the sample is fully immersed. When this relationship is applied, we can measure a weight of 79.31g of water and which is equivalent to 79.31 ml, also equal to a volume of 79 310 mm³. This volume is more accurately measured, and changes the difference of volume calculated numerically. Before there was a 15% volume overestimation from the numerical counting, considering the new value, this difference decreases to 11.6%.**

L227-229: statistically speaking one data point can be considered to be an outlier, with the remaining data points distributed around the 1:1 line. As it is your correlation is heavily weighted towards that one outlier.

**Thank you for this comment on the outlier point present in the linear relationship. We tested the same relationship without this outlier point, in order to estimate the weight of this point. The result on the seven other points exhibit a relationship of (r=0.78, n=7; p-value=0.0038) which is still satisfactory and close to the previous correlation. We added this information into the revised manuscript (l-226-227).**